# *Fox-TTS*: Scalable Flow Transformers for Expressive Zero-Shot Text to Speech

## Abstract

Expressive zero-shot text-to-speech (TTS) synthesis aims at synthesizing high-fidelity speech that closely mimics a brief stylized recording without additional training. Despite the advancements in this area, several challenges persist: 1) Current methods, which encompass implicit prompt engineering through in-context learning or by using pre-trained speaker identification models, often struggle to fully capture the acoustic characteristics of the stylized speaker; 2) Attaining high-fidelity voice cloning for a stylized speaker typically requires large amounts of specific data for fine-tuning; 3) There is no benchmark tailored for the expressive zero-shot TTS scenarios. To address them, we present *Fox-TTS*, a family of large-scale models for high-quality expressive zero-shot TTS. We introduce an improved flow-matching Transformer model coupled with a novel learnable speaker encoder. Within the speaker encoder, we incorporate three key designs: temporal mean pooling, temporal data augmentation, and an information bottleneck used for trading off pronunciation stability and speaker similarity in an explainable manner. Moreover, we have collected *Fox-eval*, the first multi-speaker, multi-style benchmark that is specially designed for expressive zero-shot scenarios. Extensive experiments show that Fox-TTS achieves on-par quality with human recordings in normal scenarios and state-of-the-art performance in expressive scenarios. Audio samples are available at `https://fox-tts.github.io/`.

## 1 Introduction

The last decade has witnessed significant progress in text-to-speech synthesis through the development of neural networks and computing resources. Currently, most TTS systems (Li et al., 2019; Shen et al., 2018; Ren et al., 2019; Kharitonov et al., 2023; Wang et al., 2023; Jiang et al., 2024b; Du et al., 2024; Anastassiou et al., 2024) adopt the cascade pipeline with an acoustic model and a vocoder (Kong et al., 2020; Lee et al., 2022) by taking mel spectrograms or acoustic tokens (Zeghidour et al., 2021; Défossez et al., 2022; Kumar et al., 2023) as the intermediate representations. Traditional TTS models (Li et al., 2019; Ren et al., 2019; 2020) excel at producing high-quality speech for known speakers using clean recording data. However, they struggle to generalize to new, unseen speakers in a zero-shot manner. To overcome this limitation, many large-scale TTS models (Wang et al., 2023; Du et al., 2024; Anastassiou et al., 2024) have emerged recently, leveraging extensive internet-sourced data to improve their generalizability and speech naturalness.

Balancing the trade-off between generation quality and speed, many large-scale TTS models have adopted a two-stage modeling pipeline that integrates autoregressive (AR) and non-autoregressive (NAR) components. For example, VALL-E 1 (Wang et al., 2023)/2 (Chen et al., 2024) initially generates the first codec code sequence in an AR manner and then fills in the remaining codes based on the preceding sequences in an NAR manner. SPEAR-TTS (Kharitonov et al., 2023) generates the semantic tokens in an AR manner and transforms them into acoustic tokens in an NAR manner. Besides, Seed-TTS (Anastassiou et al., 2024) and CosyVoice (Du et al., 2024) incorporate diffusion models in the NAR phase to enhance generation performance. However, this pipeline relies heavily on the AR part and thus suffers from slow inference speed. In contrast, other studies opt for fully NAR modeling to expedite the generation process. NaturalSpeech 2 (Shen et al., 2023) and NaturalSpeech 3 (Ju et al., 2024) employ multiple diffusion models to independently capture various acoustic characteristics. Yet, unlike these, more NAR works (Le et al., 2024; Vyas et al., 2023; Popov et al., 2021; Guan et al., 2024) only take single diffusion model based on stochastic

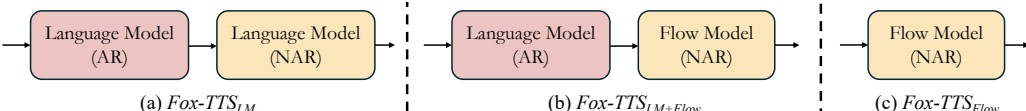

Figure 1: Three Variants of Fox-TTS.

differential equations (SDEs) (Ho et al., 2020; Song et al., 2020) or ordinary differential equations (ODEs) (Lipman et al., 2022) for speech modeling. Nonetheless, NAR models often necessitate a phoneme-level duration predictor, which can result in suboptimal prosody and increased annotation challenges when attempting to scale training data. While Seed-TTS introduces an NAR model with a sentence-level duration predictor, it offers very limited technical details.

To achieve expressive zero-shot TTS, there are mainly three strategies adopted by existing large-scale TTS models: in-context learning or utilization of a pre-trained speaker encoder or training a speaker encoder with labeled data. The strategy of in-context learning involves prompt engineering during the inference phase, where the reference speech is concatenated to the primary sequence, with the anticipation that the generated sequence will adhere to the style of the prompt (Wang et al., 2023; Chen et al., 2024; Le et al., 2024). However, our empirical observations indicate that this implicit method often falls short of accurately mimicking the stylized prompt and may be susceptible to data bias. Specifically, the in-context learning approach occasionally produces a common style that is prevalent in the training data, rather than the style presented in the given prompt. An alternative approach is to leverage a speaker encoder for explicit conditioning. Some studies (Du et al., 2024) utilize pre-trained speaker identification models to extract speaker embeddings, focusing on general timbre features while neglecting other acoustic characteristics. Other works (Jiang et al., 2024a; Guo et al., 2024) suggest learning speaker embeddings from speech data annotated with speaker labels, a process that is not only time-consuming but can not be generalized to large-scale unlabeled data.

In this work, we propose Fox-TTS, a family of large-scale models designed for highly expressive speech synthesis. As shown in Figure 1, the Fox-TTS family comprises three variants: Fox-TTS$_{LM}$, Fox-TTS$_{LM+Flow}$, and Fox-TTS$_{Flow}$. Fox-TTS$_{LM}$ is built upon the foundation of the VALL-E framework (Wang et al., 2023), enhanced with an improved Transformer architecture (Touvron et al., 2023b). However, our empirical evaluations reveal that this variant underperforms in expressive zero-shot TTS tasks, particularly concerning speaker similarity. This limitation has directed our focus towards developing a flow-matching NAR model, aimed at enhancing the versatility of speaker cloning capabilities, resulting in the other two variants: Fox-TTS$_{LM+Flow}$ and Fox-TTS$_{Flow}$. The former augments the NAR component of Fox-TTS$_{LM}$ with our proposed flow-matching model, while the latter employs solely the proposed flow-matching model[1].

Compared to existing NAR models, Fox-TTS exhibits two distinct advantages: 1) It utilizes a sentence-level duration predictor, offering greater flexibility to automatically adjust the phoneme and pause durations for improved prosody; 2) It is a fully end-to-end learnable acoustic model, independent of pre-trained models or annotated style labels such as speaker or emotion. Each component within Fox-TTS is designed to be learnable, allowing the derivation of acoustic features like timbre, prosody, and pitch from a vast array of human speech data. Formally, Fox-TTS harnesses continuous normalizing flows (CNFs) to model the transformation from a simple distribution, such as a Gaussian distribution, to a complex structured data distribution, such as human speech, conditioned on text prompts and stylized reference speeches. Given the complexity of directly optimizing CNFs, we adopt the conditional flow matching (CFM) algorithm (Lipman et al., 2022) to facilitate efficient and scalable training via a vector field regression loss.

In terms of model design, Fox-TTS contains three components: a sentence-level duration predictor, a flow-based speech denoiser, and conditional modules tailored for text and reference speech inputs. These components are constructed on an enhanced Transformer architecture, which draws inspiration from the Diffusion Transformer (DiT) (Peebles & Xie, 2023) and Llama architecture (Touvron et al., 2023b). It is noteworthy that Fox-TTS is trained on large-scale speech data crawled from the Internet, utilizing only transcripts without additional annotations. To ensure that the generated samples are faithfully consistent with the reference speech prompt, we propose a novel speaker encoder with three designs: 1) We conduct temporal data augmentation on the reference speech before forwarding the speaker encoder, which includes random clip and shuffle; 2) We employ tem-

---

[1] We slightly abuse **Fox-TTS** to denote the proposed flow Transformer model in the following paragraphs.

poral pooling on the speaker representations to mitigate semantic leakage and then introduce this time-invariant information into the flow-based denoiser through adaptive layer normalization; 3) We propose a bottleneck block to manage the representation space for explainable control. By adjusting the bottleneck dimension, Fox-TTS effectively balances pronunciation stability with speaker similarity. The proposed speaker encoder offers several advantages over previous works: 1) Compared to (Jiang et al., 2024b; Guo et al., 2024), Fox-TTS does not require speaker labels and thus can be trained at scale; 2) Compared to (Du et al., 2024), Fox-TTS does not rely on pre-trained speaker identification models but facilitates joint learning across conditional encoders and the flow-based denoiser; 3) Compared to (Chen et al., 2024; Wang et al., 2023; Le et al., 2024), Fox-TTS achieves zero-shot TTS with explicit speaker modeling rather than implicit prompt engineering through in-context learning, leading to more flexible controlling ability and faster inference speed, attributed to the shorter sequence length.

In experiments, We collect millions of hours of speech data crawled from the Internet and obtain the corresponding transcriptions through a powerful automatic speech recognition (ASR) model. Additionally, we collect a challenging benchmark **Fox-eval**, specifically designed for assessing the capabilities of expressive zero-shot TTS systems. This benchmark comprises 5,000 test samples, featuring the voices of more than 122 unique speakers across a spectrum of 10 diverse speaker domains, including settings such as outdoor interviews, TV shows, and cartoons. Extensive experiments demonstrate that Fox-TTS outperforms the state-of-the-art TTS system in expressive speaking scenarios. Notably, Fox-TTS also achieves a quality of speech that is indistinguishable from that of human recordings in normal speaking scenarios.

## 2 FOX-TTS

### 2.1 PRELIMINARY: CONDITIONAL FLOW MATCHING

**Continuous Normalizing Flows (CNFs).** Let $x = (x^1, ..., x^d) \in \mathbb{R}^d$ be the data points drawn from space $\mathbb{R}^d$, we can define a time-dependent probability density function $p : [0, 1] \times \mathbb{R}^d \to \mathbb{R}_{>0}$ and a time-dependent vector field $v : [0, 1] \times \mathbb{R}^d \to \mathbb{R}^d$. Then, a vector field $v_t$ constructs a time-dependent diffeomorphic map, termed as a flow $\phi : [0, 1] \times \mathbb{R}^d \to \mathbb{R}^d$, through the ordinary differential equation (ODE): $\frac{d}{dt}\phi_t(x) = v_t(\phi_t(x)$. A CNF transform a simple prior density $p_0$ (e.g., Guassian noise) to a complex one $p_1$ (e.g., real data): $p_t(x) = p_0(\phi_t^{-1}(x))\det[\partial \phi_t^{-1}(x)/\partial x]$.

**Conditional Flow Matching with Optimal Transport.** Given a target time-dependent probability density path $p_t(x)$ and a corresponding vector field $u_t(x)$, the flow matching objective to construct a path to match this target probability path can be defined as: $\mathcal{L}_{FM}(\theta) = \mathbb{E}_{t,p_t(x)}||v_t(x) - u_t(x)||^2$, where $v_t$ is a CNF vector field parameterized by $\theta$, $t \sim \mathcal{U}[0, 1]$, and $x \sim p_t(x)$. However, since we have no prior knowledge for $p_t$ and $u_t$, it requires an appropriate method to aggregate the the probability paths and vector fields defined for each sample. Given a particular data sample $x_1$ sampled from distribution $q$, Lipman et al. (2022) proposes that a target probability path $p_t(x)$ can be constructed via a mixture of simpler conditional probability paths $p_t(x|x_1)$. A conditional probability path is defined to satisfy $p_0(x|x_1) = p(x)$ at $t = 0$, and $p_1(x|x1)$ to be a distribution around $x = x_1$ at $t = 1$ like $\mathcal{N}(x|x_1, \sigma^2 I)$, a normal distribution with a sufficiently small standard deviation $\sigma$. Based on it, the conditional flow matching objective can be written as: $\mathcal{L}_{CFM}(\theta) = \mathbb{E}_{t,q(x_1),p_t(x|x_1)}||v_t(x) - u_t(x|x_1)||$, where $t \sim \mathcal{U}[0, 1]$, $x_1 \sim q(x_1)$, and now $x \sim p_t(x|x_1)$. Here, $u_t(x|x_1)$ can be easily computed by sampling from $p_t(x|x_1)$ since they are defined on a per-sample basis. The gradient of CFM objective w.r.t. $\theta$ is proven to be identical to the FM's. By defining the mean and the standard deviation to change linearly in time, the conditional probability flow based on the optimal transport is $p_t(x|x_1) = \mathcal{N}(x|tx_1, (1 - (1 - \sigma_{min})t)^2 I)$, where $\sigma_{min}$ is a small standard deviation. Based on it, the CFM objective with optimal transport is formulated as:

$$\mathcal{L}_{CFM-OT}(\theta) = \mathbb{E}_{t,q(x_1),p(x_0)}||\big(v_t(1 - (1 - \sigma_{min})t)x_0 + tx_1\big) - \big(x_1 - (1 - \sigma_{min})x_0\big)||^2. \quad (1)$$

### 2.2 MODEL DESIGN

Let $\mathcal{D} = (X, Y)$ be the transcribed speech dataset, where $x = (x^1, ..., x^{T_{sp}}) \in X$ denotes a speech sample of $T_{sp}$ frames and $y = (y^1, ..., y^{T_{tx}})$ denotes the corresponding transcript of $T_{tx}$ words, respectively. The goal of zero-shot TTS is to learn a mapping function $M : x = \mathcal{M}(y, x_r)$, where

Figure 2: An overview of the proposed flow-matching model Fox-TTS. The symbol "P" represents the mean pooling operation. The flow-based denoiser is subject to conditioning in two distinct yet complementary ways: First, it temporally interfaces with the phoneme sequences via a cross-attention mechanism, ensuring that the temporal dynamics of the input text are effectively captured. Second, it gets the global conditional signals through adaptive layer normalization (AdaLN), in which all the external conditions are fused.

$x_r$ is the reference speech sample. Instead of modeling the raw waveform directly, we use the mel spectrogram as the intermediate representation $z = (z^1, ..., z^{T_{mel}}) \in \mathbb{R}^{T_{mel} \times C_{mel}}$, and then decode it to the waveform with a vocoder model. Generally speaking, Fox-TTS contains a phoneme encoder, a speaker encoder, a flow-based denoiser, and a duration predictor. For each module, we use the same improved Transformer block but different parameters. Below, we initially present the improved Transformer block and then proceed to detail the design intricacies of each module. An overview of the proposed Fox-TTS architecture is presented in Figure 2.

**Improved Transformer Block.** The Transformer architecture has become a cornerstone in the realm of large language models and diffusion models. In our work, we have crafted an enhanced Transformer block, taking inspiration from the Llama model (Touvron et al., 2023a;b) and the Diffusion Transformers (DiTs) (Peebles & Xie, 2023). Our design enhancements include the implementation of rotary position encoding, which supplants absolute position encoding, as it is effective for the generation of lengthy sequences by capturing relative positions. To integrate external conditional signals, we employ two strategies comprising cross-attention and adaptive layer normalization (AdaLN). The cross-attention module is designed to link temporal-dependent variables, such as the text-speech pair. Conversely, the AdaLN module is used to incorporate global conditions such as the reference speaker representation. Out of simplicity and scalability, we apply this refined Transformer block across all subsequent modules in our model.

**Phoneme Encoder.** Let $a = (a^1, ..., a^{T_{ph}})$ be the phoneme sequence of $T_{ph}$ frames obtained from its text $y$, we first get the embedded phoneme sequence with a lookup table and then use the aforementioned improved Transformer block to encode it with removing the conditional modules (i.e., cross-attention): $f_{ph} = E_{ph}(a) \in \mathbb{R}^{T_{ph} \times C}$, where $f_{ph}$ denotes the phoneme embedding, $E_{ph}$ denotes the phoneme encoder, and $C$ is the dimension for conditions.

**Speaker Encoder.** Cloning voice with a brief recording is an important capability for zero-shot TTS systems. Recently, numerous approaches have been proposed to address this challenge, including leveraging in-context learning, utilizing pre-trained speaker encoders, and training speaker embeddings with labeled data. For instance, systems such as VALL-E 2 (Chen et al., 2024) and VoiceBox (Le et al., 2024) employ implicit prompt engineering through in-context learning to replicate reference speech characteristics. CosyVoice (Du et al., 2024) relies on a speaker identification model pre-trained with labeled speech data, while MegaTTS 2 (Jiang et al., 2024b) and RedFireTTS (Guo et al., 2024) propose learning speaker embeddings directly from labeled speech samples. In contrast, we advocate for explicit conditioning using a learnable speaker encoder, which can be trained without labels, thus capitalizing on the large-scale data crawled from the Internet.

In Fox-TTS, we use the target speech as the input of the speaker encoder during training and substitute it with the reference speech during inference. In other words, it requires the speaker encoder to prohibit semantic leakage. To achieve this, we propose three important designs: temporal data augmentation, temporal mean pooling, and an information bottleneck module. Firstly, we implement temporal data augmentation on the mel spectrogram before forwarding it to the speaker encoder. This involves two primary strategies: clipping and shuffling. The mel spectrogram is randomly segmented to 50% to 75% of its original length, followed by a temporal shuffle. This augmented mel spectrogram is then processed through the improved Transformer blocks to yield the intermediate speaker representation: $f_s = E_s(z') \in \mathbb{R}^{T_s \times C}$, where $f_s$ denotes the intermediate speaker representation, $E_s$ denotes the speaker encoder, $z'$ is the augmented mel spectrogram input, and $T_s$ and $C$

are the sequence length and the hidden dimension of the representation, respectively. Subsequently, we apply a mean pooling function to the encoded representation, thereby condensing the representational space to a point where semantic content becomes irrecoverable: $\overline{f_s} = m(f_s) \in \mathbb{R}^C$, where $m$ is the mean pooling operation.

While these steps significantly reduce the semantic content within the speaker representation, achieving complete removal is theoretically impossible. Consequently, we introduce a bottleneck module to meticulously adjust the dimensionality of the representation space: $f_{spk} = E_b(\overline{f_s}) \in \mathbb{R}^{C_{spk}}$, where $f_{spk}$ denotes the final speaker representation, $E_b$ denotes the bottleneck module, and $C_{spk} \leq C$ represents the constricted dimension. This refined design enables the empirical balancing of pronunciation stability and voice cloning similarity. For instance, constraining the representation space to a smaller dimension $C_{spk}$ enhances pronunciation accuracy while moderately affecting the similarity to the reference audio. It is noteworthy that the proposed learnable speaker encoder, which does not rely on speaker labels, can be effectively trained in conjunction with the flow-based denoiser on large-scale data.

**Flow-based Denoiser.** In Fox-TTS, we use conditional flow matching to learn the denoising probability path from Gaussian noise to the mel spectrogram distribution. Specifically, given a mel spectrogram input $z \in \mathbb{R}^{T_z \times C_{mel}}$, a phoneme embedding $f_{ph} \in \mathbb{R}^{T_{ph} \times C}$, a speaker embedding $f_{spk} \in \mathbb{R}^{C_{spk}}$, and a timestep $t \in [0, 1]$, we use a Transformer model stacked by the improved Transformer blocks to parameterize the vector field $v_t$. For practical implementation, we uniformly discretize the continuous timesteps into $T$ sampling points and generate the timestep embedding $f_t \in \mathbb{R}^T$ through a lookup table.

Notably, we introduce the conditional signals via cross-attention and AdaLN modules. The phoneme embeddings $f_{ph}$ are aligned with the noisy spectrogram input by the cross-attention module. Thereafter, we combine the projected speaker embedding, timestep embedding, and phoneme embedding post mean pooling to construct the global condition: $f_{AdaLN} = P(f_{spk}, f_t, f_{ph})$, where $P$ encompasses a suite of operations including projection, mean pooling, and addition. This global condition is subsequently propagated to the AdaLN layer of each Transformer block.

Following the CFM formulation, for a mel spectrogram input $z$ and a prior sample $z_0$ (e.g., a noise sampled from Gaussian distribution), we derive $z_t = zt + (1 - (1 - \sigma_{min})t)z_0$ and $u_t(z_t|z) = z - (1 - \sigma_{min})z_0$. Accordingly, the learning objective of Fox-TTS within the CFM-OT framework can be formulated as:

$$\mathcal{L}_{Fox-TTS}(\theta) = \mathbb{E}_{t,q(z,a),p_0(z_0)}||u_t(z_t|z) - v_t(z_t, z', a; \theta)||^2. \tag{2}$$

**Sentence Duration Predictor.** In Fox-TTS, we opt for sentence-level duration over phoneme-level duration to determine the sequence length during inference. Given the variability in expression modes among individuals for the same sentence, this results in diverse target speech lengths. Inspired by this variability, we input both phoneme sequences and learnable speaker embeddings into the duration predictor. The architecture of the sentence duration predictor is similar to other modules, i.e., constructed by stacking several improved Transformer blocks. We use the $L1$ regression loss to optimize the sentence duration predictor.

## 2.3 TRAINING AND INFERENCE

**Training.** The learning objective defined in Eq. (2) uniformly trains the vector field $v_t$ across all timesteps within the interval [0,1]. However, the complexity of learning varies with different timesteps. For instance, the optimal prediction at $t = 0$ is straightforwardly the mean of $p_1$, whereas the task becomes increasingly challenging as $t$ approaches the midpoint of [0,1]. Consequently, it is important to implement a weighted training loss that allocates greater emphasis on the intermediate timesteps. As discussed in (Esser et al., 2024), incorporating a time-dependent weighting does not alter the optimal solution of the learning objective. In practice, we transition the timestep sampling distribution from a uniform distribution to a logit-normal distribution (Atchison & Shen, 1980) with probability density function $\pi(t)$, which is equivalent to applying a weighted training loss:

$$\pi(t; \mu, s) = \frac{1}{s\sqrt{2\pi}} \frac{1}{t(1 - t)} \exp\left(-\left(\log \frac{t}{1 - t} - \mu\right)^2 / (2s^2)\right), \tag{3}$$

where $\mu$ denotes the location parameter and $s$ represents the scale parameter. At $\mu = 0$ and $s = 1$, the distribution conforms to the standard logit-normal distribution. In Fox-TTS, we use the standard

logit-normal distribution for timestep sampling since it reports a stable performance on text-to-image generations Esser et al. (2024). By introducing this timestep sampling technique, we empirically observe $> 2x$ convergence speed up than the variant using uniform timestep sampling.

**Inference** Given the learned parameterized vector field $v_t(z_t, z', a; \theta)$ and a noise $z_0$ sample drawn from the prior distribution $p_0$, we can approximate the target sample $\phi_1(z_0)$ using an ODE solver. This process involves estimating $v_t$ at multiple timesteps $t \in [0, 1]$ to approximate the probability flow. Generally, employing a higher number of estimation times yields a more precise solution for $\phi_1(z_0)$, albeit at the cost of increased inference complexity. For Fox-TTS, we empirically observe that utilizing 10 and 25 sampling steps is sufficient for Fox-TTS$_{\text{LM+Flow}}$ and Fox-TTS$_{\text{Flow}}$, respectively. Note that the number of sampling steps can be further reduced by many techniques like rectified flows (Liu et al., 2022; 2023).

During inference, we adopt the classifier-free guidance (CFG) to strike a balance between the diversity and fidelity of the generated samples. In the context of diffusion probability models, CFG is realized by merging the estimated conditional scores with the unconditional ones, where the unconditional model is derived by randomly dropping the conditional inputs with a certain probability. We extend this technique to Fox-TTS by adjusting the estimated vector field as follows:

$$\hat{v}_t(z', a; \theta) = \gamma v_t(z', a; \theta) + (1 - \gamma)v_t(\emptyset; \theta), \tag{4}$$

where $\gamma > 1$ is the CFG scale and $v_t(\emptyset; \theta)$ is obtained by dropping all the condition signals (i.e., phoneme and reference speech).

## 3 EXPERIMENTS

### 3.1 EXPERIMENTAL SETUP

**Datasets.** Fox-TTS is trained on a vast dataset of internet-crawled speech data. To maximize the utility of this data with minimal human intervention, we have developed an automated processing workflow, culminating in the creation of a speech dataset that spans millions of hours, termed **Fox-train**. This workflow is crafted to handle both long and short audio segments, as well as to facilitate data feature extraction. The long audio processing module standardizes audio formats, performs resampling, and segments audio files based on speaker identity and duration. The short audio processing module contains noise reduction and quality assessment. Data feature extraction encompasses the transcription of text and phonemes, along with the extraction of continuous (i.e., mel spectrogram) and discrete (i.e., Encodec) audio features. This workflow is engineered for large-scale parallel processing without the need for manual intervention and thus ensures the scalability of Fox-TTS across vast datasets.

On the other hand, to thoroughly evaluate the expressive zero-shot speech generation capabilities of our model across diverse scenarios and to demonstrate its high-quality performance, we develop a specialized test set named **Fox-eval**. This large-scale, diverse, and stylistically varied test set imposes more challenges on generating the speech with specific styles and tones than conventional test sets. Specifically, Fox-eval contains 5,000 test samples with 122 speakers from 10 distinct domains, including outdoor interviews, TV shows, and cartoons. By employing Fox-eval, we aim to conduct a comprehensive assessment of the model's performance across various speech scenarios, thereby ensuring its reliability and efficacy in practical applications. More details about Fox-eval can be found in Appendix A.1.

**Model Configuration.** We provide detailed model configurations in Appendix A.2.

**Training and Inference.** For Fox-TTS$_{\text{LM}}$, we use the Adam optimizer with a learning rate of 3.0e-4 and a batch size of 480. For Fox-TTS$_{\text{LM+Flow}}$, we reuse the AR part of Fox-TTS$_{\text{LM}}$ and train the flow-matching Transformer model by the Adam optimizer with a learning rate of 1.0e-4 and a batch size of 144. For Fox-TTS$_{\text{Flow}}$, the learning rate is set to 5.0e-5, while keeping the other settings the same as the flow-matching Transformer model in Fox-TTS$_{\text{LM+Flow}}$. For classifier-free guidance, we set the dropping probability to 0.2 during training and set the CFG scale $\gamma = 3$ during inference. After generating the target mel spectrogram, an in-house trained HiFi-GAN model (Kong et al., 2020) is applied to convert it into the waveform. All models are trained on a large-scale NVIDIA A100 cluster.

**Metrics.** We conduct objective evaluations using the Word Error Rate (WER) and Speaker Similarity (SIM) metrics. For WER, we utilize Paraformer as our automatic speech recognition (ASR) engine (Gao et al., 2023). In the context of SIM, we leverage Resemblyzer[2] to generate speaker embeddings, which are subsequently employed to compute the cosine similarity between speech samples of each test utterance and corresponding reference speech. Furthermore, we conduct Mean Opinion Score (MOS) studies for subjective evaluation. During the MOS evaluation, evaluators first listen to a reference speech clip of the target speaker. They then listen to a synthesized speech sample generated by a randomly selected model. Evaluators are asked to rate the synthesized speech on a scale from 1 to 5 based on its similarity to the reference clip in terms of prosody and expressiveness.

## 3.2 RESULTS OF ZERO-SHOT SPEECH SYNTHESIS

**Main Results.** In this subsection, we assess the zero-shot capabilities of our proposed Fox-TTS models by comparing them with existing models using the Fox-eval dataset, which includes both subjective and objective evaluations. We evaluate three variants of Fox-TTS: 1) Fox-TTS$_{LM}$, a two-stage model that uses fully language models to generate discrete speech tokens, followed by a codec-based vocoder for synthesis; 2) Fox-TTS$_{LM+Flow}$, which also uses a language model for speech token generation but then applies a diffusion model to produce spectral features; and 3) Fox-TTS$_{Flow}$, a one-stage model that directly processes text and reference speech inputs using a diffusion model, leveraging a sentence-level length predictor for audio length prediction. These models are trained on the Fox-train dataset to improve their zero-shot generalization capabilities. For comparison, the most recent work CosyVoice (Du et al., 2024) is chosen for its superior performance on zero-shot TTS tasks and open-source availability. Besides, most large-scale TTS models Chen et al. (2024); Ju et al. (2024); Anastassiou et al. (2024); Guo et al. (2024) are not released due to many reasons like security concerns. In our implementation, we utilize the publicly available pre-trained CosyVoice model[3] for evaluation.

| System | WER (↓) | SIM (↑) | MOS (↑) |
|---|---|---|---|
| CosyVoice (Du et al., 2024) | 4.16 | 0.854 | 4.03 |
| Fox-TTS$_{LM}$ | 2.01 | 0.781 | 3.83 |
| Fox-TTS$_{LM+Flow}$ | **1.58** | 0.843 | **4.12** |
| Fox-TTS$_{Flow}$ | 3.44 | **0.868** | 3.98 |

Table 1: Performance on the **Fox-eval** benchmark.

**Analysis.** As shown in Table 1, we provide a detailed assessment using both objective metrics and subjective evaluations. CosyVoice exhibits strong speaker similarity but underperforms in terms of word error rate. Subjective listening tests indicate a higher incidence of prosody errors, indicating that although the general voice timbre aligns with the target speaker, it fails to capture the subtle prosodic elements essential for expressive speech synthesis. This issue likely stems from CosyVoice's reliance on a pre-trained speaker encoder, which captures only global timbral characteristics, and its use of semantic tokens that do not adequately represent the complexity of detailed timbral features.

In the following three rows of the table, we present the objective and subjective performance metrics for three Fox-TTS variants. It is worth noting that Fox-TTS$_{LM}$ shares the same AR model with Fox-TTS$_{LM+Flow}$, but differs in its NAR implementation. Specifically, Fox-TTS$_{LM}$ employs multi-level codec predictions and introduces prompts through a prefix method. In contrast, Fox-TTS$_{LM+Flow}$ leverages a learnable speaker encoder and predicts continuous spectral features using a diffusion model. This architectural divergence results in marked enhancements in both timbre similarity and pronunciation stability for Fox-TTS$_{LM+Flow}$. As a result, Fox-TTS$_{LM+Flow}$ achieves the lowest WER and the highest subjective MOS scores among the evaluated models. In addition, Fox-TTS$_{Flow}$ excels in timbre imitation and prosody, as indicated by its superior performance on the SIM scores. This suggests that the direct mapping from text to speech provided by the diffusion model adeptly captures the subtleties of speech. These results highlight the effectiveness of the Fox-TTS models in improving both timbre similarity and pronunciation accuracy, thereby validating our designs.

---

[2]https://github.com/resemble-ai/Resemblyzer
[3]https://github.com/FunAudioLLM/CosyVoice

## 3.3 COMPARISON WITH HUMAN SPEAKERS

We compare the performance of our Fox-TTS models to that of human speakers using the DiDiSpeech dataset (Guo et al., 2021). As shown in Table 2, Fox-TTS achieves a WER lower than human speakers, indicating superior pronunciation accuracy. And the similarity score also slightly exceeds that of human speakers. Most significantly, we conduct a subjective evaluation on the synthesized audio samples and the human recordings. The Comparative Mean Opinion Score (CMOS) for Fox-TTS is nearly equivalent to that of human speech, with an infinitesimal difference of -0.05, indicating that Fox-TTS can generate speech at a human level. These results demonstrate that Fox-TTS is capable of producing speech that is not only intelligible but also natural, effectively reaching human-level quality in speech synthesis. While surpassing human recordings on objective metrics does not signify that there is no room for improvement, it is a fact that the generated audio can sometimes be accompanied by noise that leads to a decline in sound quality. This is also why there is still a small gap in the subjective evaluation.

| System | WER (↓) | SIM (↑) | CMOS (↑) |
|--------|---------|---------|----------|
| Human | 0.87 | 0.852 | - |
| Fox-TTS | **0.74** | **0.863** | -0.05 |

Table 2: Performance on the DiDiSpeech dataset.

## 3.4 DISCUSSION OF THE FOX-EVAL BENCHMARK

The general performance metrics presented in Table 1 provide a preliminary evaluation of the model's capabilities. However, it cannot fully reveal the detailed performance across various speaking scenarios. Consequently, this section extends the analysis to uncover subtleties in model behavior by examining the performance data from the Fox-eval benchmark in Table 3. Through this detailed investigation, we aim to bring to the fore critical insights that are often marginalized in zero-shot speech synthesis research.

| System / Category | Fox-TTS$_{LM}$ | | Fox-TTS$_{LM+Flow}$ | | Fox-TTS$_{Flow}$ | |
|---|---|---|---|---|---|---|
| | WER (↓) | SIM (↑) | WER (↓) | SIM (↑) | WER (↓) | SIM (↑) |
| Cartoon | 1.46 | 0.794 | 1.43 | 0.843 | 2.71 | 0.871 |
| Stylized | 1.65 | 0.825 | 1.30 | 0.870 | 3.04 | 0.896 |
| Role-Playing | 1.60 | 0.791 | 1.49 | 0.849 | 4.99 | 0.870 |
| Outdoor Interview | 1.97 | 0.801 | 1.78 | 0.823 | 3.77 | 0.845 |
| TV Show | 2.86 | 0.741 | 1.88 | 0.834 | 3.59 | 0.859 |
| Monologue | 2.32 | 0.829 | 1.14 | 0.897 | 2.22 | 0.918 |
| Casual Conversation | 2.52 | 0.771 | 1.53 | 0.848 | 3.36 | 0.872 |
| Film Actor | 2.22 | 0.743 | 2.02 | 0.805 | 4.52 | 0.842 |
| Customer Support | 1.11 | 0.811 | 0.96 | 0.840 | 2.20 | 0.886 |
| Articulate Speaker | 1.35 | 0.812 | 1.16 | 0.880 | 2.73 | 0.893 |

Table 3: Performance of each speaker category on the **Fox-eval** benchmark.

Upon examining the benchmark data, it is evident that models display a proficiency in scenarios where speech is highly structured and contains minimal variability. For instance, the models achieve notably lower WER and higher SIM scores in categories such as Customer Support and Articulate Speaker. This trend can be attributed to the standardized and deliberate manner in which speech is delivered in these contexts. On the other hand, categories demanding greater expressiveness or variability, such as Role-Playing and Film Actor, present more significant challenges. The discrepancy between model performance in expressive scenarios is often overlooked in previous zero-shot evaluations, including those conducted with LibriSpeech (Panayotov et al., 2015) and DiDiSpeech (Guo et al., 2021), among others. This oversight is a key driver for the development of the Fox-eval benchmark. Both subjective and objective experiments consistently demonstrate the exceptional suitability of the Fox-eval benchmark for assessing zero-shot models in high-performance scenarios. The comprehensive nature of the benchmark ensures thorough evaluation, revealing that Fox-TTS maintains a superior level of performance across both general and high-expressiveness contexts, outperforming other zero-shot models. This consistent performance underscores the superiority of the approach.

### 3.5 ABLATION STUDY

In this section, we explore the critical design elements of the speaker encoder within our Fox-TTS model and conduct ablation studies to elucidate their individual impacts on model performance. Specifically, there are three key designs: temporal data augmentation, temporal mean pooling, and the information bottleneck module. The results of these ablation studies are shown in Table 4.

| System | WER ($\downarrow$) | SIM ($\uparrow$) |
|---|---|---|
| Fox-TTS | 1.58 | 0.843 |
| w/o temporal mean pooling | - | - |
| w/o temporal data augmentation | 1.90 | 0.832 |
| w/o information bottleneck | 1.71 | 0.848 |

Table 4: Ablation studies of the proposed speaker encoder.

Specifically, the absence of temporal mean pooling leads to aberrant outcome, highlighting its crucial role in the speaker encoder. The exclusion of temporal data augmentation is associated with a significant increase in WER and a decrease in SIM, mainly due to the reduced ability to mitigate the leakage of content information from the reference audio within the learnable speaker encoder. Moreover, the removal of the information bottleneck is primarily marked by an increase in WER. While eliminating the bottleneck might offer a slight improvement in SIM, it also leads to a decline in pronunciation stability and audio quality. Collectively, these findings affirm the essential design of the speaker encoder as vital for achieving superior performance in zero-shot speech synthesis.

### 3.6 HYPER-PARAMETER SELECTION

In the inference stage, two hyper-parameters (i.e., the number of ODE steps and the CFG scale $\gamma$) are important to the quality of the generated samples. We analyze the selection of these hyper-parameters with the Fox-TTS$_{\text{LM+Flow}}$ in Table 5. We can observe that using 10 ODE sampling steps is a good choice to balance generation quality and inference speed. Similar experiments are also conducted for Fox-TTS$_{\text{Flow}}$, in which we find using 25 ODE sampling steps is a balanced choice. For the classifier-free guidance scale $\gamma$, we study the values ranging from 1.0 to 5.0 with an interval of 1.0. From the listed results, we can find that the CFG is important for improving the generation quality, especially for the WER metric, and set the CFG scale $\gamma = 3$ is the best choice.

| ODE Steps | WER ($\downarrow$) | SIM ($\uparrow$) | CFG scale $\gamma$ | WER ($\downarrow$) | SIM ($\uparrow$) |
|---|---|---|---|---|---|
| 3 | 3.12 | 0.748 | 1.0 | 1.09 | 0.862 |
| 5 | 1.07 | 0.848 | 2.0 | 0.79 | 0.864 |
| 10 | 0.72 | 0.860 | 3.0 | 0.74 | 0.863 |
| 15 | 0.76 | 0.862 | 4.0 | 0.75 | 0.859 |
| 25 | 0.74 | 0.863 | 5.0 | 0.77 | 0.854 |

Table 5: The results of Fox-TTS with different hyper-parameter settings. We set the CFG scale $\gamma = 3$ when evaluating the effect of the number of ODE steps, and 25 ODE steps when evaluating the effect of the CFG scale $\gamma$.

## 4 CONCLUSION

In this paper, we introduce Fox-TTS, a family of high-quality zero-shot text-to-speech models. Considering that language modeling has been extensively studied, there is a lack of comprehensive research on diffusion or flow-matching models for large-scale TTS training. Therefore, we propose a flow Transformer model with several novel designs to enable large-scale training on unlabeled data. In experiments, to address the absence of a tailored benchmark in the field of zero-shot TTS, we collect a multi-speaker, multi-style dataset called Fox-eval. Experiments on Fox-eval and DiDiSpeech demonstrate that Fox-TTS achieves the state-of-the-art performance and is comparable to human recordings. In future work, we will continue to enhance the generation quality, particularly for the efficient Fox-TTS$_{\text{Flow}}$ model, and develop watermarking techniques to ensure proper use.

BROADER IMPACTS

Since Fox-TTS can synthesize highly realistic speech with a short reference recording, it may carry out some potential risks in misusing, such as spoofing voice identification or synthesizing harmful content with a specific speaker. We continue to build systems to prevent these situations by developing fake audio detection models and audio watermarking techniques. We plan to release our pre-trained models after strict security checks.

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

# Appendices

# *Fox-TTS*: Scalable Flow Transformers for Expressive Zero-Shot Text to Speech

## A EXPERIMENTAL DETAILS

### A.1 TESTSET

We collect a multi-style and multi-speaker dataset for evaluating zero-shot TTS systems. Concretely, there are 122 different speakers of 10 different types, including cartoon, stylized, role-playing, outdoor noisy conversation, TV show, monologue, casual conversation, film actor, customer support, and articulate speaker. Besides, the textual contents are collected from 8 resources, including novels, game lines, dictionaries, legal books, WeChat public accounts, exams, encyclopedias, and dialogue. The proportion of speaker types and content resources are illustrated in Figure 3.

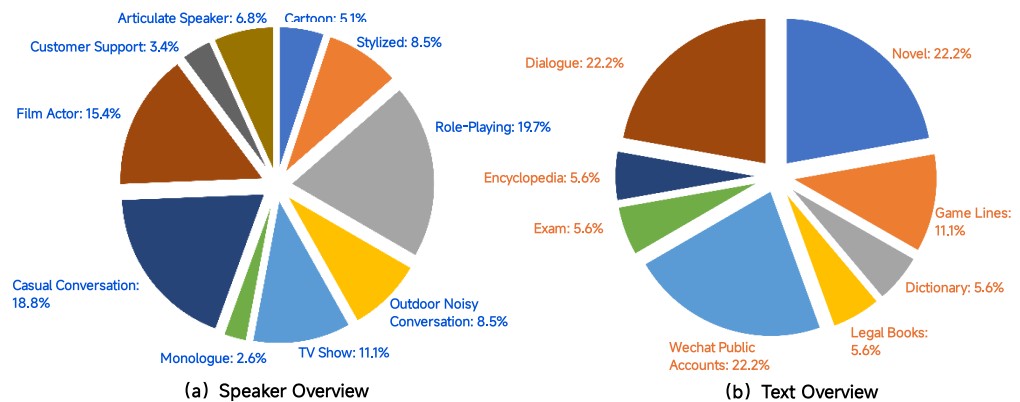

Figure 3: An overview of the proposed testset **Fox-eval**.

### A.2 MODEL CONFIGURATION

The proposed flow-based Transformers are applied to two variants: Fox-TTS$_{LM+Flow}$ and Fox-TTS$_{Flow}$. Compared to Fox-TTS$_{Flow}$, Fox-TTS$_{LM+Flow}$ has an additional token encoder module, which is built upon the improved Transformer block. We provide detailed hyper-parameter settings about Fox-TTS$_{LM+Flow}$ and Fox-TTS$_{Flow}$ in Table 6 and 7, respectively. Additionally, the hyper-parameter of the vocoder are also shown in Table 8.

| Model Hyper-parameter | | Fox-TTS$_{LM+Flow}$ |
|---|---|---|
| Phoneme Encoder | Encoder Layers | 3 |
| | Phoneme Embedding Size | 250 |
| | Hidden Size | 1024 |
| | Max Sequence Length | 1500 |
| Token Encoder | Encoder Layers | 3 |
| | Token Embedding Size | 1030 |
| | Hidden Size | 1024 |
| | Max Sequence Length | 3000 |
| Speaker Encoder | Encoder Layers | 1 |
| | Hidden Size | 1024 |
| | Max Sequence Length | 3000 |
| Flow-based Denoiser | Dncoder Layers | 16 |
| | Number of Attention Heads | 32 |
| | Hidden Size | 768 |
| | Max Sequence Length | 3000 |
| | Discretized Flow Timesteps | 1000 |
| Total Params | | 334M |

Table 6: Model configurations for the flow transformer of Fox-TTS$_{LM+Flow}$

| Model Hyper-parameter | | Fox-TTS$_{Flow}$ |
|---|---|---|
| Phoneme Encoder | Encoder Layers | 6 |
| | Phoneme Embedding Size | 250 |
| | Hidden Size | 1024 |
| | Max Sequence Length | 1500 |
| Speaker Encoder | Encoder Layers | 2 |
| | Hidden Size | 1024 |
| | Max Sequence Length | 3000 |
| Flow-based Denoiser | Dncoder Layers | 16 |
| | Number of Attention Heads | 32 |
| | Hidden Size | 1152 |
| | Max Sequence Length | 3000 |
| | Discretized Flow Timesteps | 1000 |
| Total Params | | 684M |

Table 7: Model configurations for Fox-TTS$_{Flow}$

| Model Hyper-parameter | | Vocoder |
|---|---|---|
| Generator | Upsample Rates | [8,4,4,2] |
| | Upsample Kernel Sizes | [16,8,8,4] |
| | Upsample Initial Channel | 1024 |
| | Resblock Kernel Sizes | [3,7,11] |
| | Resblock Dilation Sizes | [[1,3,5], [1,3,5], [1,3,5]] |
| Total Params | | 54M |

Table 8: Model configurations for vocoder

