# OpenReview forum: "Fox-TTS: Scalable Flow Transformers for Expressive Zero-Shot Text to Speech"
_ICLR.cc/2025/Conference — ICLR 2025 Conference Withdrawn Submission_

### Official Review · Reviewer_8owX · 2024-10-18

**Soundness:** 1
**Presentation:** 2
**Contribution:** 2
**Rating:** 3
**Confidence:** 4

**Summary:**

This work introduces a flow-matching Transformer model coupled with a learnable speaker encoder for expressive zero-shot TTS. They also introduce a benchmark/dataset for expressive TTS.

It presents solid implementation details, but I find some confusing claims regarding motivation and experiments:
- It mentions previous works (vall-e, cosyvoice, etc.) with prompt mechanism or speaker encoder **struggle to fully capture the acoustic characteristics of the stylized speaker**,  but actually I tried these open-sourced models and find them with strong ability to follow different prompt speakers, the reason that they don't perform well on stylized speaker is more possibly due to their less stylized training data (that could also be why the authors propose a new dataset)
- The author attributes the problem to speaker encoder, but their own design is also not related to "speaker", what they do is data augmentation, temporal mean pooling, bottleneck and keep it learnable during training process. I wonder how can these simple/engineering designs improve the speaker encoder's representations?
- The author lists the weakness of many previous TTS methods (line 198 - 206), but their experiment results only compare to cosyvoice.

**Strengths:**

see sbove

**Weaknesses:**

see sbove

**Questions:**

I find one claim for bottleneck (line 224-227): **This refined design enables the empirical balancing of pronunciation stability and voice cloning similarity. For instance, constraining the representation space to a smaller dimension C_spk enhances pronunciation accuracy while moderately affecting the similarity to the reference audio.**

How can you obtain such conclusion and where are your experiment results to verify it?

---

### Official Review · Reviewer_w6b3 · 2024-10-29

**Soundness:** 2
**Presentation:** 2
**Contribution:** 1
**Rating:** 3
**Confidence:** 4

**Summary:**

This paper presents an expressive zero-shot text-to-speech model using a large-scale expressive dataset. The authors highlight the advantages of an learnable explicit speaker encoder for speaker conditioning methods while recent models utilized in-context prompting methods.

**Strengths:**

1. The authors show the potential of learnable speaker encoder as an alternative method to in-context prompting.

2. They present Fox-eval benchmark for expressive zero-shot scenarios.

**Weaknesses:**

1. While this paper presents a new benchmark for zero-shot text-to-speech, they only compare the model with CozyVoice. I recommend adding further experiments with LibriSpeech or Seed-TTS eval or DiTTo-TTS eval. Please refer F5-TTS paper and their evaluation methods.
Compared to this paper, F5-TTS made an effort to provide a more comprehensive evaluation. (https://github.com/SWivid/F5-TTS)

2. It seems to be a limited contribution from a research perspective without presenting a new benchmark. The model architecture is almost the same as Seed-TTS. The only difference is that this paper utilizes a speaker encoder. However, using a speaker encoder is not novel. Furthermore, while the authors stated that the limitation of in-context prompt for speaker conditioning methods, they do not compare the different speaker encoding methods. An ablation study on different speaker conditioning methods would enhance the study's impact.

3. Additionally, there are three model variations: Fox-TTS_LM, Fox-TTS_LM+Flow, Fox-TTS_Flow. However, they only trained the model with a speaker encoder conditioning. Each model would exhibit different tendencies depending on the conditioning method used. Without experimental evidence, the authors seem to have drawn conclusions somewhat hastily.

4. The authors stated that “there is a lack of comprehensive research on diffusion or flow-matching models for large-scale TTS training.” However, TorToise, VoiceBox, DiTTo-TTS, Seed-TTS, E2-TTS, OpenVoice, and CozyVoice have already provided insights into large-scale TTS training.

**Questions:**

Please add English audio samples to the demo page.

[Q1] Please add WavLM-based SIM evaluation for a similarity metrics to demonstrate the effectiveness of speaker encoding method compared to in-context prompting.

---

### Official Review · Reviewer_39t4 · 2024-11-04

**Soundness:** 1
**Presentation:** 2
**Contribution:** 2
**Rating:** 3
**Confidence:** 4

**Summary:**

In contrast to recent trends in TTS that use variable length speaker embeddings for providing speaker information to speech synthesis models, this paper claims that it is possible to achieve state-of-the-art results in WER and Speaker Similarity by using fixed length speaker embedding obtained through mean-pooling. The paper proposes providing speaker information through the adaptive layer norm of the transformer parameterizing the text and speaker conditional generative model that is optimized with the optimal transform conditional flow matching objective.

Though the authors also mention an automated pipeline for large scale TTS data ingestion and an evaluation dataset for expressive TTS, it is my perception, given the lack of detail regarding these, that the authors do not position te dataset and the dataset creation pipeline as the main contribution of the paper.

**Strengths:**

The authors propose an approach for speaker conditioning that differs from existing approaches in terms of implementation. That is, the authors condition the model with a speaker embedding provided through the adaptive layer norm layer of a transformer. Though similar to existing  approaches in TTS and Voice Conversion, the novelty lies on conditioning through Adaptive Layer Norm versus other modules.

The authors march against the current trend that shows that variable length speaker embeddings are superior to fixed length speaker embeddings. This is laudable but comes at the cost of being received with skepticism and requiring careful comparison with the current preferred method.

The authors also mention a large scale dataset obtained with an automated pipeline for data ingestion, and an evaluation dataset for expressive speech.

**Weaknesses:**

From a dataset perspective, it is unclear that the dataset or the pipeline to produce the dataset used by the authors will be shared with the community. Details about the dataset and its pipeline are also lacking.

From the perspective of the proposed approach, the results in the paper are controversial, given that replacing fixed-length speaker embeddings with variable length speaker embeddings became the de-facto choice for achieving state-of-the-art zero shot speaker similarity in TTS. This has been show in works such as SpeechFlow, VoiceBox, P-Flow and, more recently, E2TTS.

I find it surprising that, without justification, the authors use an experimental setup that is quite different from the setup described in the models using the approach they claimed to be superior to, such as Vall-E, VoiceBox, P-Flow and more recently E2TTS.

In this context, it is impossible to accurately assess the efficiency of the proposed approach, and to position it with respect to the approaches they claim to surpass.

**Questions:**

In the paper you mention using mean pooling and Figure 2 shows multiple "boxes" as the output of the Speaker Encoder. Throughout my assessment I'm assuming that this mean pooling operation is over time and produces a fixed length vector given a variable length input to produce f_spk.

The data augmentation you are using on the data going into the speaker encoded has been used before by works in TTS and Voice Conversion. Please cite them accordingly.

I urge the authors to follow the experimental setup in the Vall-E, VoiceBox, P-Flow, and others. It consists of providing WER, SIM-O and SIM-R on sentences from LibriSpeech Test Clean with durations between 4 and 10 seconds. While doing such, I urge the authors to use the same speech transcription model and speaker embedding model used in these works.

I also urge the authors to be mindful of their choice of transcriptions for this evaluation, given that it can have negative impact on WER. A reasonable choice can be found here https://www.openslr.org/145

Finally, given that data can have a significant impact on speaker similarity in zero-shot TTS, I urge the authors to follow the aforementioned experimental setup and benchmark the efficiency of their method in different data regimes. Specifically, I urge the authors to follow the data regimes in P-Flow (LibriTTS) or SpeechFlow and (LibriLight) VoiceBox, such that one can separately measure the impact of the data and the impact of the proposed approach, and more directly compare with existing approaches.

Last, the paper uses the word "semantic" but does not define what it refers to. Note that F0 contours and timbre, for example, are semantic: they carry information about emotion, sex, accent. If in the manuscript the word "semantic" refers to phonemes, which I suspect they do, please use the term phonemes, otherwise define what it refers to.

Though I am unwilling to support this work in its current state, I'm happy to support it if the authors run the experiments above and show, through this new evaluation, including ablation on different data regimes, that their approach is at least comparable to the approaches they claim to superior.

---

### Official Review · Reviewer_h5Wx · 2024-11-04

**Soundness:** 2
**Presentation:** 2
**Contribution:** 1
**Rating:** 3
**Confidence:** 4

**Summary:**

This paper proposes Fox-TTS, a flow-matching-based model for zero-shot TTS, capable of multi-speaker and multi-style zero-shot TTS by using a learnable speaker encoder. The speaker encoder incorporates temporal mean pooling, temporal data augmentation, and an information bottleneck. The authors claim that it achieves performance similar to human recordings.

**Strengths:**

* It outperforms the baseline, CosyVoice, in terms of pronunciation accuracy (WER) and speaker similarity (SIM).

* On the DiDiSpeech dataset, the paper shows that Fox-TTS achieves objective metric values similar to those of human recordings.

* For various scenarios, it demonstrates good results in pronunciation accuracy and speaker similarity across the three proposed methods."

**Weaknesses:**

* The paper only compares with a single baseline, CosyVoice. There have been numerous zero-shot TTS models, but this paper only compares against CosyVoice, which uses a speaker encoder, as its baseline. Since zero-shot TTS approaches utilizing in-context learning, such as VALL-E or VoiceBox, have shown better speaker similarity than speaker encoder-based methods. However, this study relies solely on a speaker encoder for speaker adaptation. To demonstrate the advantages of this approach, recent zero-shot TTS models like VoiceBox, NaturalSpeech 2, and 3 should be used as baselines for comparison.

* In the paper, Figure 1 introduces Fox-TTS_LM, Fox-TTS_{LM+Flow}, and Fox-TTS_Flow as variants of the proposed Fox-TTS, but the model description details only Fox-TTS_Flow, lacking detailed explanations of the other two variants.

* All samples on the demo page are in Chinese, making it difficult for readers who do not speak Chinese to evaluate the model's performance.

**Questions:**

* Exactly how much data was used to train the model?

* Also, what languages make up the training data?

* What impact did using the logistic normal distribution have on the final performance?"

---

### Note · Authors · 2024-12-24

I have read and agree with the venue's withdrawal policy on behalf of myself and my co-authors.